# Multiple Instance Learning with Trainable Soft Decision Tree Ensembles

Andrei Konstantinov [†] [iD], Lev Utkin [†] [iD] and Vladimir Muliukha *,[†] [iD]

Department of Artificial Intelligence, Peter the Great St.Petersburg Polytechnic University, Polytechnicheskaya, 29, 195251 St. Petersburg, Russia; konstantinov_av@spbstu.ru (A.K.); utkin_lv@spbstu.ru (L.U.)
* Correspondence: vladimir.muliukha@spbstu.ru
† These authors contributed equally to this work.

**Abstract:** A new random forest-based model for solving the Multiple Instance Learning problem under small tabular data, called the Soft Tree Ensemble Multiple Instance Learning, is proposed. A new type of soft decision trees is considered, which is similar to the well-known soft oblique trees, but with a smaller number of trainable parameters. In order to train the trees, it is proposed to convert them into neural networks of a specific form, which approximate the tree functions. It is also proposed to aggregate the instance and bag embeddings (output vectors) by using the attention mechanism. The whole Soft Tree Ensemble Multiple Instance Learning model, including soft decision trees, neural networks, the attention mechanism and a classifier, is trained in an end-to-end manner. Numerical experiments with well-known real tabular datasets show that the proposed model can outperform many existing multiple instance learning models. A code implementing the model is publicly available.

**Keywords:** multiple instance learning; decision tree; oblique tree; random forest; attention mechanism; neural network





## 1. Introduction

Many machine learning real-life applications deal with labeled objects called bags, which consist of several instances wherein individual labels of the instances contained in the bags are not provided. For example, in histopathology, a histology image can be viewed as a bag and its patches (cells) as instances of the bag [1–3]. One can find many similar examples of applications, such as drug activity prediction [4], detecting lung cancer [5], protein function annotation [6], etc. A useful framework for modeling the above applications is Multiple Instance Learning (MIL), which can be regarded as a kind of weakly supervised learning [7–13]. The MIL objectives are, firstly, to classify new bags, based on training data consisting of a set of labeled bags, and, secondly, to classify unlabeled instances in the bags. In order to achieve the objectives, assumptions or rules are introduced to establish relationships between labels of instances and labels of the corresponding bag. Most MIL models assume that all negative bags contain only negative instances, and that positive bags contain at least one positive instance. However, there are also other rules regarding bag label definitions [14].

There are many MIL models which try to solve the classification problem under different conditions and for different types of datasets [15–20]. Most of the above models use such methods as the support vector machine, K nearest neighbors, convolutional neural networks, and decision trees. An interesting and efficient class of the MIL models applies the attention mechanism [21–27].

However, there are drawbacks of the above approaches to MIL. On the one hand, simple models, based on methods like the support vector machine and decision trees, do not use neural networks and cannot attain the advantages of the network models; for

example, the end-to-end training and the attention mechanism. On the other hand, the MIL models based on neural networks cannot be accurately trained on small tabular datasets.

Therefore, the MIL model which simultaneously has properties of the random forest (RF) and the neural network can provide better results in comparison with the available model. The RF directly fits an MIL model because it is robust to noise in target variables. At the same time, its structure is suboptimal; therefore, it does not minimize the MIL loss in some cases. One of the ways to construct decision trees, which can be retrained, is by means of the concept of *soft oblique trees* [28], the trainable parameters of which can be updated and optimized by using gradient-based algorithms. Oblique trees, and RFs composed of oblique trees, use linear and non-linear classifiers at each split in the decision trees and allow the combining of more than one feature at a time. However, when we deal with *small tabular data*, soft oblique trees may overfit, due to a large number of parameters.

Therefore, we propose to represent soft oblique trees in the form of classic decision trees and to convert the decision trees, which comprise the RF, into trainable neural networks in a special way. The corresponding neural networks implement approximately the same functions as the decision trees, but they can be effectively trained jointly with the attention mechanism and can simultaneously take into account data from all bags, i.e., they successfully solve the MIL problem.

As a result, we propose an attention MIL model, called Soft Tree Ensemble MIL (STE-MIL). On the one hand, STE-MIL is based on decision trees and successfully deals with small tabular data. On the other hand, after converting trees into neural networks and applying the attention mechanism to aggregate embeddings of instances and bags, STE-MIL is trained by using gradient descent algorithms in an end-to-end manner.

Our contributions can be summarized as follows:

1. A new RF neural network-based MIL model is proposed, which outperforms many existing MIL models when dealing with small tabular data.
2. A new type of soft decision trees, similar to the soft oblique trees, is proposed. In contrast to the soft oblique trees, the proposed trees have a smaller number of trainable parameters. Nevertheless, the soft decision trees can be trained in the same way as the soft oblique trees. Outputs of each soft decision tree are viewed as a set of vectors (embeddings) which are formed from the class probability distributions in a specific way.
3. An original algorithm for converting the decision trees into neural networks of a specific form for efficiently training parameters of the trees is proposed.
4. The attention mechanism is proposed to aggregate the instance and bag embeddings with the aim of minimizing the corresponding loss function.
5. The whole MIL model, including the soft decision trees, neural networks, the attention mechanism and a classifier, is trained in an end-to-end manner.
6. Numerical experiments with the well-known datasets, Musk1, Musk2 [4], Fox, Tiger, and Elephant [15] illustrate STE-MIL. The above datasets have numerical features that are used to perform tabular data. The corresponding code implementing STE-MIL is publicly available at https://github.com/andruekonst/ste_mil (accessed on 17 July 2023).

The paper is organized as follows. Related work is available in Section 2. An introduction to MIL and the oblique binary soft trees is given in Section 3. A specific representation of the decision tree function, which allows us to convert the decision tree to a neural network, is proposed in Section 4. A soft tree ensemble to solve the MIL problem is considered in Section 5. An algorithm for converting the decision trees into neural networks is studied in the same section. The attention mechanism applied to the proposed MIL models is studied in Section 6. Numerical experiments are described in Section 7. Section 8 considers open research questions, provides some discussion, and offers concluding remarks.

## 2. Related Work

**MIL**. MIL can be regarded as an important tool for dealing with different types of data. In particular, tabular data of a specific structure can be classified by means of MIL models.

When considering tabular data, several available MIL models are based on applying such models as SVM, decision trees, AdaBoost, and RFs [15,16,19,20,29].

However, most MIL models are based on applying neural networks or convolutional neural network, and especially so when image datasets are classified [17–19,30–33].

In spite of many available MIL models, there are no models which could combine tabular data-oriented models, such as RFs and neural networks, including the attention mechanism, in order to use the gradient-based algorithm for updating training parameters of RFs, as well as neural networks, and to improve accuracy in the MIL predictions.

**MIL and attention**. Several MIL models using the attention mechanism have been proposed in order to enhance classification accuracy, examples of the models are the following: SA-AbMILP (Self-Attention Attention-based MIL Pooling) [34], ProtoMIL (Multiple Instance Learning with Prototypical Parts) [26], MHAttnSurv (Multi-Head Attention for Survival Prediction) [24], AbDMIL [23], MILL (Multiple Instance Learning-based Landslide classification) [35], DSMIL (Dual-Stream Multiple Instance Learning) [36]. The attention-based MIL models can also be found in [21,22,27,37,38]. The main peculiarity of the above-mentioned models is that they use neural networks and mainly deal with image data rather than small tabular data.

**Oblique trees and neural networks**. Many studies have demonstrated that trees with oblique splits, in many cases produce smaller trees with better accuracy than that with axis parallel trees [39,40]. One of the important advantages of oblique trees is that they can be trained by using optimization algorithms, an example being the gradient descent algorithm. On the other hand, some obstacles can be encountered in training oblique trees. In particular, the training procedure is computationally expensive. Moreover, the corresponding model may be overfitted. Several approaches have been proposed to partially solve the above problems. Wickramarachchi et al. [40] presented a new decision tree algorithm, called HHCART. In order to simplify oblique trees, Carreira-Perpinan and Tavallali [41] proposed an algorithm called sparse oblique trees, which produces a new tree from the initial oblique tree having the same or smaller structure, but new parameter values leading to a lower or unchanged misclassification error. One-Stage Tree, as a soft tree to build and prune the decision tree jointly through a bi-level optimization problem, is presented in [42]. Menze et al. [43] focused on trees with task-optimal recursive partitioning. Katuwal et al. [44] proposed a random forest of heterogeneous oblique decision trees that employ several linear classifiers at each non-leaf node on some top-ranked partitions. An application of evolutionary algorithms to the problem of oblique decision tree induction is considered in [45]. An algorithm improving learning of trees through end-to-end training with backpropagation was presented in [46].

An interesting direction of using oblique trees is in representing neural networks as the trees, or the trees in the form of neural networks. Lee et al. [47] showed how neural models can be used to realize piece-wise constant functions, such as decision trees. Hazimeh et al. [48] proposed combining the advantages of neural networks and tree ensembles in designing a hybrid model by considering the so-called tree ensemble layer for neural networks, which is an additive model of differentiable decision trees. The layer can be inserted anywhere in a neural network, and is trained along with the rest of the network using gradient-based algorithms. Frosst and Hinton [49] took the knowledge acquired by a neural net and expressed the same knowledge in a model that relies on hierarchical decisions instead, so that explaining a particular decision would be much easier. A way of using a trained neural net to create a type of soft decision tree that generalizes better than one learned directly from the training data was provided in [49]. Karthikeyan et al. [50] proposed a unified method that enables accurate end-to-end gradient-based tree training and can be deployed in a variety of settings. Madaan et al. [51] presented dense gradient trees and a transformer, based on the trees, which is called Treeformer.

In contrast to the above work, we considered how to apply decision trees to the MIL problem by converting the trees into neural networks of a special form and by training them jointly with the attention mechanism in an end-to-end manner.

## 3. Preliminary

### 3.1. Multiple Instance Learning

First, we formulated the MIL classification problem [7–9,13]. It differs from the standard classification in data structure. Namely, in the MIL problem, bags have class labels, but instances, which compose each bag, are usually unlabeled. As a result, this problem can be regarded as a kind of weakly supervised learning problem. Due to the availability of labels which are only for bags, the following tasks can be stated in the framework of MIL. The first task is concerned with annotation of instances from a bag. The second task aims to annotate new bags by having a training set of bags, i.e., the task is to train a classifier at the bag level. The above tasks can be solved by introducing special rules which establish the relationship between instances and the bag class labels.

Let us formally state the MIL problem taking into account the rules connecting different levels of the MIL data consideration. Suppose that each bag is defined by a set of $n$ instances $\mathbf{X} = \{\mathbf{x}_1, \ldots, \mathbf{x}_n\}$, where the $i$-th instance $\mathbf{x}_i \in \mathbb{R}^m$ is represented by the feature vector. Each instance $\mathbf{x}_i$ has a label $y_i \in \{0, 1\}$ taking two values: 0 (negative class) and 1 (positive class). We do not know labels $y_i$ during training, as it follows from the MIL problem statement. According to the first task, we constructed a function $g$ which maps each vector $\mathbf{x}_i$ into the label $y_i$.

There are various rules establishing the relationship between labels of bags and instances. One of the most common rules can be rewritten as follows: [9]:

$$f(\mathbf{X}) = \begin{cases} 1, & \exists \mathbf{x} \in \mathbf{X} : g(\mathbf{x}) = 1, \\ 0, & \text{otherwise,} \end{cases} \tag{1}$$

where $f(\mathbf{X})$ is a bag classifier.

It follows from (1) that at least one positive instance makes the bag positive, and negative bags contain only negative instances. As an example, we considered histopathological images divided into several patches [12,14]. The whole image with one of the labels "cancer" or "non-cancer" can be viewed as a bag, whereas each patch of the image can be regarded as an instance. The function $f(\mathbf{X})$, taking values 1 and 0, corresponds to the image labels "cancer" or "non-cancer", respectively. The function $g(\mathbf{x})$, which also takes values 1 and 0, corresponds to the patch labels "cancer" or "non-cancer", respectively. If the image is from the high-risk patient, then it should be labeled as "cancer" if at least one of all patches belonging to the image contain malignant tumor [12].

On the other hand, if the patient is low risk, then the rule establishing the relationship between labels of bags and instances can be relaxed, i.e., some number of "cancer" patches are necessary to assign the "cancer" label to the whole histopathological image. In this case, the function $f(\mathbf{X})$ can be defined in another way, taking into account a threshold $\theta$; for example, the number of "cancer" patches, can be defined as

$$f(\mathbf{X}) = \begin{cases} 1, & \theta \leq \sum_{\mathbf{x} \in \mathbf{X}} g(\mathbf{x}), \\ 0, & \text{otherwise.} \end{cases} \tag{2}$$

We use the rule defined in (1).

The dataset can be represented as

$$\mathcal{D} = \left\{ \left( \{\mathbf{x}_k^{(i)}\}_{k=1}^{n_i}, y_i \right) \right\}_{i=1}^{N}, \tag{3}$$

where $\mathbf{x}_k^{(i)}$ is the $k$-th instance vector belonging to the $i$-th bag; $n_i$ is the number of instances in the $i$-th bag; $N$ is the number of labeled bags in the training set.

Rule (1), defining the function $f$, can be represented through the MIL maximal pooling operator as follows:

$$f(\{\mathbf{x}_k^{(i)}\}_{k=1}^{n_i}) = \max\left\{g(\mathbf{x}_k^{(i)})\right\}_{k=1}^{n_i}. \tag{4}$$

Hence, the binary classification loss, which is minimized, can be written as:

$$\mathcal{L} = \frac{1}{N}\sum_{i=1}^{N} l\left(y_i, f(\{\mathbf{x}_k^{(i)}\}_{k=1}^{n_i})\right) = \frac{1}{N}\sum_{i=1}^{N} l\left(y_i, \max\{g(\mathbf{x}_k^{(i)})\}_{k=1}^{n_i}\right). \tag{5}$$

### 3.2. Oblique Binary Soft Trees

One of the important procedures to build oblique decision trees is optimization of their parameters. There are various decision rules for building trees. The so-called hard decision rules have been successfully implemented in [50,51]. The rules are applied to oblique decision trees which may be improper when we deal with small tabular datasets because large degrees of freedom, in this case, would lead to overfitting.

According to [50], an oblique binary tree of a height $h$ represents a piece-wise constant function $f(\mathbf{x}; \mathbf{W}, \mathbf{b}) : \mathbb{R}^m \to \mathbb{R}^K$, parameterized by weights $\mathbf{w}_{I(d,l)} \in \mathbb{R}^m$, $b_{I(d,l)} \in \mathbb{R}$ at a node on the path from the tree root to its leaf $l$ at the depth $d$. Here, $I(d,l)$ is the index of a node on the path from the tree root to its leaf $l$ with depth $d$. Function $f$ computes decision functions of the form $\mathbf{w}_j^{\mathrm{T}}\mathbf{x} - b_j > 0$, that define whether $\mathbf{x}$ must traverse the left or right child next. Here $\mathbf{W}$ is the parameter matrix consisting of all parameter vectors $\mathbf{w}_j$; $\mathbf{b}$ is the parameter vector consisting of parameters $b_j$. The tree output is represented as $2^h$ vectors $\theta_1, \ldots, \theta_{2^h}$ such that vector $\theta_j \in \Delta^K$ at the $j$-th leaf is associated with probabilities of $K$ classes, where $\geqq^K$ is the unit simplex of dimension $K$. One of the ways to learn parameters $\mathbf{w}_{ij}$ and $b_{ij}$ for all nodes is to minimize the expected loss $l$ of the form:

$$\min_{\mathbf{W},\mathbf{b}} \sum_{i=1}^{n} l(y_i, f(\mathbf{x}; \mathbf{W}, \mathbf{b})). \tag{6}$$

Karthikeyan et al. [50] propose the following function $f_\theta(\mathbf{x}; \mathbf{W}, \mathbf{b})$:

$$f_\theta(\mathbf{x}; \mathbf{W}, \mathbf{b}) = \sum_{l=1}^{2^h} q_l(\mathbf{x}, \mathbf{W}, \mathbf{b}) \cdot \theta_l, \tag{7}$$

where the tree path indicators $q_l(\mathbf{x}, \mathbf{W}, \mathbf{b})$ are represented as the following indicator functions:

$$q_l(\mathbf{x}, \mathbf{W}, \mathbf{b}) = \mathbb{I}\left[\bigwedge_{d=1}^{h} ([\mathbf{w}_{I(d,l)}^{\mathrm{T}}\mathbf{x} \leq b_{I(d,l)}] \oplus s(d,l))\right], \tag{8}$$

Here, $\mathbb{I}(\cdot)$ is the indicator function taking the value 1 if its argument is non-negative, and otherwise it is 0; $\oplus$ is the operator *XOR*; $s(d,l)$ determines whether the predicate of a node on the path to the leaf $l$ at the depth $d$ should be evaluated to be true or false, i.e., $s(d,l) = 1$ if the $l$-th leaf belongs to the left subtree of node $I(d,l)$, otherwise $s(d,l) = 0$. It is well known that the conjunction in (8) can be replaced with the product, as follows:

$$q_l(\mathbf{x}, \mathbf{W}, \mathbf{b}) = \prod_{d=1}^{h} \mathbb{I}\left[[\mathbf{w}_{I(d,l)}^{\mathrm{T}}\mathbf{x} \leq b_{I(d,l)}] \oplus s(d,l)\right]. \tag{9}$$

However, this representation significantly complicates the optimization of the model by using the gradient descent algorithm, due to the vanishing gradient problem. Another representation of $q_l(\mathbf{x}, \mathbf{W}, \mathbf{b})$ is proposed in [50]. It is of the form:

$$q_l(\mathbf{x}, \mathbf{W}, \mathbf{b}) = \sigma\left(\sum_{d=1}^{h} \sigma\left[[\mathbf{w}_{I(d,l)}^{\mathrm{T}}\mathbf{x} \leq b_{I(d,l)}] \oplus s(d,l)\right] - h\right), \tag{10}$$

where the indicator functions are replaced with the so-called $\sigma$-hard indicator approximations [50], which apply quantized functions in the forward pass, but use the smooth

activation functions in the backward pass to propagate. This specific representation of the sigmoid function is called the straight-through operator and is proposed in [52].

The above representation allows us to effectively apply the gradient descent algorithm to compute optimal parameters of the tree in accordance with the loss function (6).

The soft tree concept proposed in [50,51] is an interesting approach to deal with small tabular data. However, our experiments with soft trees have demonstrated that it is difficult to train oblique soft trees for many datasets. Therefore, we proposed modifying the standard decision trees so as to implement them in the form of neural networks.

## 4. A Softmax Representation of the Decision Tree Function

In order to overcome difficulties of training the oblique decision tree, we propose another representation of it, which allows us to effectively update it. Let us consider a complete binary decision tree $f_\theta$ of depth $h$:

- the tree has $(2^h - 1)$ non-leaf nodes parametrized by $(\mathbf{w}_j, b_j)$, where

  - $\mathbf{w}_j$ is an *one-hot vector* having 1 at the position corresponding to the node feature;
  - $b_j$ is a threshold;

- the tree also has $2^h$ leaves with values $\mathbf{v}_l$, where $\mathbf{v}_l$ is an output vector corresponding to the $j$-th leaf.

In contrast to the representation (10) of the function $q_l$, we propose avoiding direct comparison with the height of a tree, because this representation requires the indicator approximation to return integer values; otherwise $q_l$ is always evaluated as zero. If we use (10) instead of the softmax function, then (7) provides the sum of the leaf vectors in place of selecting one of them. We use the softmax function to guarantee a convex combination of leaf vectors. We replace the outer indicator with the *softmax* function having the trainable temperature parameter $\tau$:

$$q_l(\mathbf{x}, \mathbf{W}, \mathbf{b}, \tau, \omega) = \operatorname{softmax}_\tau \left( \sum_{d=1}^{h} \sigma_\omega \left[ \left[ -\mathbf{w}_{I(d,l)}^{\mathrm{T}} \mathbf{x} + b_{I(d,k)} \right] \cdot \hat{s}(d,k) \right] \right)_{k=1}^{2^h}, \qquad (11)$$

where $\hat{s}(d,k) \in \{-1, 1\}$ is the node sign; $\sigma$ is the sigmoid with the trainable temperature or scaling parameter $\omega$.

The proposed representation could be interpreted as selecting the most appropriate path among all candidate paths. Neural trees, defined by using the above representation, can be optimized by means of the stochastic gradient descent algorithm with *fixed node weights* $\mathbf{w}_j$; i.e., by updating only thresholds, the softmax temperature parameters $\tau$, the sigmoid temperature parameters $\omega$, and the leaf values.

## 5. Soft Tree Ensemble for MIL

One of the possible ways for solving the MIL classification problem, i.e., for constructing the instance model $\tilde{g}$, is to assign a bag label to all instances belonging to the bag. In this case, we obtain a new instance-level dataset with the repeated instance labels, which is of the form:

$$\tilde{\mathcal{D}} = \{ \left( \mathbf{x}_k^{(i)}, y_i \right) \mid k = 1, \ldots, n_i \}_{i=1}^{N}. \qquad (12)$$

According to [53], the RF can be regarded as a desirable MIL classifier, even if it is trained on artificially made instance-level datasets, such as (12), because the RF is inherently robust to noise in the target variable. After training on the dataset (12), parameters of the built RF can be seen as a suboptimal solution to the optimization problem defined by the bag-level loss (5). In the extreme worst case scenario, the RF is totally overfitted, i.e., it just remembers the bag label for each instance.

There are approaches that try to repeatedly infer the instance labels by using the trained RF, and then retrain the RF on the obtained instance labels. One such approach is implemented in the so-called MIForests [53]. The main problem of the results is that the

methods rebuild decision trees instead of updating them, partially losing the useful tree structures obtained at different steps.

*5.1. Soft Tree Ensemble*

A key idea behind STE-MIL can be represented in the form of the following schematic algorithm:

1. Let us assign incorrect labels to instances of a bag; for example assigning the same label as that of the corresponding bag. The instance labels may be incorrect because we do not know true labels and their determination is our task. However, these labels are needed to build an initial RF. This is a kind of initialization procedure for the whole model, which is trained in the end-to-end manner.
2. The next step is to convert the initial RF to a neural network having a specific architecture. To implement this step, non-leaf nodes of each tree in the RF are parametrized by trainable parameters $\mathbf{b}$, $\tau$, $\omega$, and non-trainable parameters $\mathbf{W}$.
3. Parameters of the tree nodes $\mathbf{b}$, $\tau$, $\omega$ are updated by using the stochastic gradient descend algorithm to minimize the bag loss defined in (5). To implement the updating algorithm, we propose approximating the tree path indicators $q_l(\mathbf{x}, \mathbf{W}, \mathbf{b}, \tau, \omega)$, by using the specific softmax representation (11). This is a key step of the algorithm which allows us to update trees by updating neural networks and incorporating trees or the RF in the whole scheme of modules, including the attention mechanism and a classifier.

Suppose that the RF consisting of $T$ decision trees has been trained on the repeated instance labels (12). We convert its trees to a set of $T$ neural networks which implement functions $f^{(1)}(\mathbf{x}), \ldots, f^{(T)}(\mathbf{x})$, such that the $i$-th tree corresponds to the $i$-th network implementing the function $f^{(i)}(\mathbf{x})$. After converting trees to neural networks, we can update their parameters to minimize bag-level loss (5). The ensemble prediction for a new instance $\mathbf{x}$ is defined as follows:

$$f(\mathbf{x}) = \frac{1}{T} \sum_{i=1}^{T} f^{(i)}(\mathbf{x}). \tag{13}$$

The bag prediction can be obtained by applying any aggregation function $G$:

$$f(\{\mathbf{x}_k^{(i)}\}_{i=1}^{N}) = G(f(\mathbf{x}_1^{(i)}), \ldots, f(\mathbf{x}_{n_i}^{(i)})). \tag{14}$$

The next question is how to convert the decision trees into neural networks.

*5.2. Trees to Neural Networks*

Suppose that the RF is trained on the artificial dataset (12). Then, it can be converted to a neural network with a specific structure. A tree with $M$ internal decision nodes and $L$ leaves is represented as a neural network with the following three layers:

1. The first layer aims to approximate the node predicates. It is a fully connected layer with $m$ inputs (dimensionality of $\mathbf{x}$) and $M$ outputs, i.e., it is held that:

$$f^{(1)}(\mathbf{x}) = \sigma(\mathbf{W}\mathbf{x} + \mathbf{b} \mid \omega), \tag{15}$$

where $\mathbf{W} \in \mathbb{R}^{r \times m}$ is a matrix of non-trainable parameters consisting of $r$ vectors $\mathbf{w}_i \in \mathbb{R}^m$; $r$ is the total number of the tree nodes; $\mathbf{b} \in \mathbb{R}^r$ is the trainable bias vector; $\omega$ is the trainable temperature parameter of the sigmoid $\sigma$.

As a result, the first layer has only trainable parameters $\mathbf{b}$ and $\omega$. The matrix $\mathbf{W}$ consists of one-hot vectors having 1 at positions corresponding to the node features.

2. The second layer aims to estimate the leaf indices. It is fully connected layer with $M$ inputs and $L$ outputs having one trainable parameter $\tau$:

$$f^{(2)}(\xi) = \text{softmax}(\mathbf{R}\xi + \mathbf{s} \mid \tau), \tag{16}$$

where $\mathbf{R} \in \mathbb{R}^{L \times M}$ is a non-trainable routing matrix that encodes decision paths, such that one path forms one row of $\mathbf{R}$; $\mathbf{R}\xi \in \mathbb{R}^M$ is the input vector; $\mathbf{s} \in \mathbb{R}^L$ is the non-trainable bias vector; $\tau$ is the trainable temperature parameter of the softmax operation.

Matrix $\mathbf{R}$ consists of values from the set: $\{-1, 0, 1\}$. If the path to $i$-th leaf does not contain $j$-th node, then $R_{i,j} = 0$. Otherwise, if the path goes to the left branch, then $R_{i,j} = -1$, and $R_{i,j} = 1$ if the path goes to the right branch. The vector $\mathbf{s} = (s_1, \dots, s_{2^h})$ is needed to balance the decision paths. The sum of the sigmoid functions from the path to the $k$-th leaf in (11) can be represented as:

$$\sum_{d=1}^{h} \sigma\Big( [-\mathbf{w}_{I(d,l)}^{\mathrm{T}} \mathbf{x} + b_{I(d,k)}] \cdot \hat{s}(d,k) \Big)$$
$$= \sum_{i=1}^{M} \Big( R_{k,i} \sigma\Big( -\mathbf{w}_{I(d,l)}^{\mathrm{T}} \mathbf{x} + b_{I(d,k)} \Big) + \mathbb{I}[R_{k,i} = -1] \Big)$$
$$= \sum_{i=1}^{M} \Big( R_{k,i} \sigma\Big( -\mathbf{w}_{I(d,l)}^{\mathrm{T}} \mathbf{x} + b_{I(d,k)} \Big) \Big) + s_k, \tag{17}$$

because it holds that $\sigma(-\omega) = 1 - \sigma(\omega)$.

3.  The third layer aims to calculate the output values (embeddings). It is trainable and fully connected. Each leaf generates the class probability vector of the size $C$. We take the probability $v_1(\mathbf{x})$ of class 1 and repeat it $E - 1$ times, such that the whole embedding $\mathbf{v}(\mathbf{x}) = (v_1^{(1)}(\mathbf{x}), \dots, v_1^{(E)}(\mathbf{x}))$ has the length $E$. The final output of the network (or the third layer) is of the form

$$f(\mathbf{x}) = \mathbf{V} f^{(2)}\Big( f^{(1)}(\mathbf{x}) \Big), \tag{18}$$

where $\mathbf{V} \in \mathbb{R}^{E \times L}$ is a trainable leaf value matrix consisting of $L$ vectors $\mathbf{v}(\mathbf{x})$.

An example of the transformation of a tree to a neural network is illustrated in Figure 1. A full decision tree with three decision nodes and four leaves was considered and is depicted in Figure 1. The first layer of the neural network computes all decisions at internal nodes of the tree. Matrix $\mathbf{R}$ is constructed, such that each row represents a path to the corresponding leaf of the tree. For example, values of the first row are $(-1, -1, 0)$ because the path to the leaf $l_1$ passes through the nodes $d_1$ and $d_2$ to the left. Values of the third row are $(1, 0, -1)$ because the path to the third leaf $l_3$ passes through the node $d_1$ to the right, and does not pass through the node $d_2$ and passes through the node $d_3$ to the left. Elements of the vector $\mathbf{s}$ are equal to the number of left turns, which is equivalent to the number of values $-1$ at the corresponding row of $\mathbf{R}$.

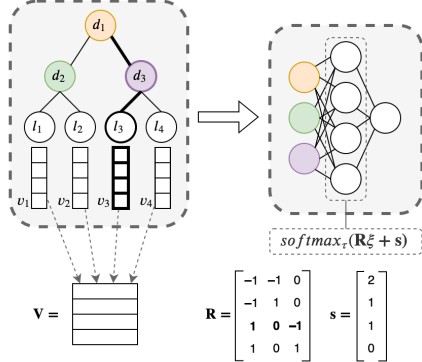

**Figure 1.** Forming the second layer through matrix $\mathbf{R}$.

The class distribution provided by a tree is computed by counting the percentage of different classes of instances at the leaf node into which the concerned instance falls.

Formally, the leaf value vectors are initially estimated for the *l*-th leaf of the *j*-tree as follows:

$$\mathbf{v}_j^{(l)}(\mathbf{x}) = \left( \frac{\#\{(k,i) \in J(l) | y_i = 1\}}{\#J(l)} \right)_{t=1}^{E}, \tag{19}$$

where $J(l)$ is an index set of training points which fall into the *l*-th leaf.

We used constant matrix $\mathbf{W}$, representing decision nodes, in order to preserve the axis parallel decision planes. It is initialized as the one-hot encoded representation of decision tree split features. Only the bias $\mathbf{b}$ of the first layer of the neural network is trainable, and is initialized with the negative values of the decision tree split thresholds.

Matrix $\mathbf{V}$ of the leaf values is initialized with repeated tree leaf values, i.e., each column contains the same values equal to the original tree leaf value.

The algorithm for the routing matrix $\mathbf{R}$ construction is shown as Algorithm 1.

---

**Algorithm 1** Recursive $\mathbf{R}$ matrix construction

---

   **procedure** FILL($\mathbf{R}$, $a$, $b$, $k$)
      **if** $k > M$ **then return**
      **end if**
      $d \leftarrow \lfloor \frac{b-a}{2} \rfloor$                                             ▷ A half of the input row index span $[a, b]$
      **for** $i \leftarrow a$ to $a + d$ **do**
         $R_{i,k} \leftarrow -1$                                       ▷ Fill first $d$ rows with $-1$
         $R_{i,d+k} \leftarrow 1$                                    ▷ Fill second $d$ rows with $1$
      **end for**
      FILL($\mathbf{R}$, $a$, $a + d$, $2k$)                     ▷ Recursively fill the first $d$ rows, left subtree
      FILL($\mathbf{R}$, $a + d$, $b$, $2k + 1$)                  ▷ Fill the second $d$ rows, right subtree
   **end procedure**
   $\mathbf{R} \leftarrow \mathbf{0} \in \mathbb{R}^{L \times M}$                                   ▷ Initialize matrix with zeros
   FILL($\mathbf{R}$, $a = 1$, $b = L$, $k = 1$)

---

### 5.3. Peculiarities of the Proposed Soft Trees

- The sigmoid and softmax temperature parameters are trained starting from value 0.1 to avoid having to fit them as hyperparameters. Temperatures as trainable parameters are not redundant because the first layer of the neural network contains a fixed weight matrix $\mathbf{W}$, so $\mathbf{Wx} + \mathbf{b}$ cannot be equivalent to $\tau(\mathbf{Wx} + \mathbf{b})$. The same takes place with the softmax operation, which contains a fixed number of terms from 0 to 1.
- In contrast to [50], we did not use oblique trees, as they may lead to overfitting on tabular data. Trees with the axis-parallel separating hyperplanes allow us to build accurate models for tabular data where linear combinations of features often do not make sense.
- Therefore, we also did not use overparametrization, which is a key element for convergence in training the decision trees with quantized decision rules (when the indicator is represented not by a sigmoid function, but by the so-called straight-through operator [52]).
- We used softmax as an approximation of the argmax operation instead of the approximation of the sum of indicator functions. At the prediction stage, the implementation of the algorithm proposed in [50], which uses the sigmoid function, could predict the sum of the values at several leaves at the same time.

Further, we can reduce the temperature $\omega$ so that the decision rules become more stringent. Unfortunately, this does not work in practice because, by a rather large depth ($h > 3$), on the same path, inconsistent rules are often learned, which give the "correct" values by low temperatures and degenerate by small $\omega$. As a result, accuracy starts to decrease as $\omega$ decreases. If we do not decrease $\omega$, then the trees may no longer be axis-parallel.

## 6. Attention and the Whole Scheme of STE-MIL

After training, the output of each neural network corresponding to the $k$-th tree is the embedding $\mathbf{v}_{j,k}^{(i)}$ of length $E$, where $i$ and $j$ are indices of the corresponding bag and instance in the bag, respectively. This implies that we obtain $T$ embeddings $\mathbf{v}_{j,1}^{(i)}, \ldots, \mathbf{v}_{j,T}^{(i)}$ for the $j$-th instance from the $i$-th bag, $j = 1, \ldots, n$, $i = 1, \ldots, N$, under an assumption of an identical number of trees in all RFs. It should be noted that numbers of trees in RFs can be different. However, we considered the same numbers for simplicity.

Embeddings $\mathbf{v}_{i,1}, \ldots, \mathbf{v}_{i,T}$ are aggregated by using, for example, the averaging operation, resulting in vectors $\mathbf{e}_j^{(i)}$, $j = 1, \ldots, n$, corresponding to the $i$-th bag. Then, aggregated embeddings $\mathbf{e}_1^{(i)}, \ldots, \mathbf{e}_n^{(i)}$ are attended to in order to obtain a final representation of the $i$-th bag in the form of vector $\mathbf{a}_i$, which is classified. This motivated us to replace the class probability distributions at the tree leaves with the embeddings $\mathbf{v}$ defined above. We can define several ways to construct embeddings from class probability distributions. However, we selected a simple procedure, which has demonstrated its efficiency from the point of view of accuracy and computationally.

Hence, the second idea behind STE-MIL is to aggregate the embeddings over all bags by using the attention mechanism and to calculate the prediction logits by the linear projection of the aggregated embedding to the one-dimensional space. This idea is also motivated by the Attention–MIL approach proposed in [23], and by the Multi-attention multiple instance learning model proposed in [25], which may help to train a better bag-level classifier. A scheme of the whole STE-MIL model is shown in Figure 2. It can be seen from Figure 2 that each instance ($\mathbf{x}_j^{(i)}$) from the $i$-th bag learns the corresponding RF such that embeddings are combined to the aggregated vector $\mathbf{e}_j^{(i)}$. Vectors $\mathbf{e}_j^{(i)}$, $j = 1, \ldots, n_i$, can be regarded as keys in terms of the attention mechanism. They attend to and produce vector $\mathbf{a}_i$, which is the input of the classifier. The whole system is trained on all instances from all bags.

The attention module produces a new aggregate embedding $\mathbf{a}_k$ corresponding to the $k$-th bag, which is computed as follows:

$$\mathbf{a}_k = \sum_{i=1}^{n} \beta_i^{(k)} \mathbf{e}_i^{(k)}, \; k = 1, \ldots, N, \tag{20}$$

where

$$\beta_i^{(k)} = \text{softmax}\left(\mathbf{q}^{\mathrm{T}} \mathbf{k}_i\right), \tag{21}$$

$$\mathbf{q} = \mathbf{V}_q \mathbf{g}, \; \mathbf{k}_i = \mathbf{V}_k \mathbf{e}_i^{(k)}. \tag{22}$$

Here $\mathbf{V}_k$ and $\mathbf{V}_q$ are the trainable weight matrices for $\mathbf{e}_i^{(k)}$ (keys) and the template vector $\mathbf{g}$ (query), respectively.

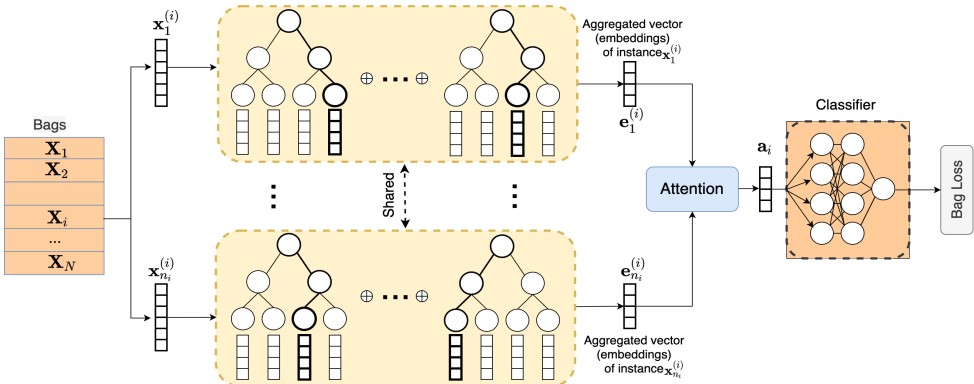

**Figure 2.** A scheme of the ensembled STE-MIL.

### 7. Numerical Experiments

In order to compare the proposed model with other existing MIL classification models, we trained the corresponding models on the well-known datasets Musk1, Musk2 (drug activity) [4], Fox, Tiger, and Elephant (images divided into patches) [15]. Table 1 shows the number of bags $N$, the number of instances $n$ in every bag and the number of features $m$ in instances for the corresponding datasets. The Musk1 dataset contains 92 bags, such that each bag consists of 476 instances having 166 features. The average bag size is 5.17. The Musk2 dataset contains 102 bags such that each bag consists of 6598 instances with 166 features. The average bag size is 64.69. Each dataset (Fox, Tiger and Elephant) contains exactly 200 bags consisting of different numbers of instances with 230 features. Numbers of instances in datasets Fox, Tiger and Elephant are 1302, 1220 and 1391, respectively. The average bag sizes of the datasets are 6.60, 6.96 and 6.10, respectively.

**Table 1.** A brief introduction to datasets for classification.

| Data Set | $N$ | $n$ | $m$ |
|---|---|---|---|
| Elephant | 200 | 1391 | 230 |
| Fox | 200 | 1302 | 230 |
| Tiger | 200 | 1220 | 230 |
| Musk1 | 92 | 476 | 166 |
| Musk2 | 102 | 6598 | 166 |

The considered datasets were created on the basis of practical tasks. In particular, datasets Musk1 and Musk2 are real drug activity prediction datasets, which allowed us to study how a molecule possesses a musky nature. The task using these databases was to classify molecules as positive (musk) or negative (non-musk) [4]. A molecule exists in multiple conformations (its certain configurations), which can be regarded as unlabeled instances in the MIL terminology, because it is not possible to observe and measure characteristics of a single conformation [9]. At the same time, one can observe the characteristics for the corresponding molecule. Hence, the molecule is labeled positive if one or more of its conformations show muskiness. It can be regarded as a bag.

Datasets Fox, Tiger, and Elephant are obtained from images which contain the corresponding animals [15]. Each image is divided into small patches represented by color, texture and shape features. If at least one patch in the image contains an animal, then the image, as a bag, is labeled positive.

The accuracy measures for these datasets were also obtained by means of the following well-known MIL classification models; mi-SVM [15], MI-SVM [15], MI-Kernel [54], EM-DD [55], mi-Graph [56], miVLAD [57], miFV [57], mi-Net [19], MI-Net [19], MI-Net with DS [19], MI-Net with RC [19], Attention and Gated-Attention [23].

We investigated Extremely Randomized Trees (ERT) for initialization because they provide better results. At each node, the ERT algorithm chooses a split point randomly for each feature and then selects the best split among these [58].

In the experiments we also used the following: sigmoid function with the trainable temperature parameter $\omega$, initialized with 10, as an indicator approximation; softmax operation with the trainable temperature parameter $\tau$, which was also initialized with 10; the number $T$ of decision trees was 20; the largest depth $h$ of trees was 5; the dimension $E$ of each embedding vector was 4; the number of epochs was 2000; the batch size was 20 and the learning rate was 0.01.

Accuracy measures (the mean and standard deviations) are computed by using five-fold cross-validation. The best results are in bold in tables. Numerical results for datasets Elephant, Fox and Tiger are shown in Table 2. It can be seen from Table 2 that STE-MIL outperformed on all datasets. Numerical results for datasets Musk1 and Musk2 are shown in Table 3. One can see from Table 3 that the proposed model outperformed all the other models for the dataset Musk1. However, STE-MIL provided the worst result for the dataset

Musk2. One of the reasons for this result was that bags in Musk2 consisted of many instances. This implies that the advantage of STE-MIL to deal with small datasets cannot be shown on this dataset.

It should also be pointed out that the values of standard deviations estimated for the Attention [23] and Gated-Attention [23] models were smaller than standard deviations corresponding to the STE-MIL results for all considered datasets (see Tables 2 and 3). One of the reasons was that the number of decision trees $T = 20$ taken in numerical experiments with STE-MIL was rather small. On the one hand, if we increase this number, then the number of training parameters also increases, leading to overfitting. On the other hand, the small number of trees can be a reason for uncertainty of results. Reducing uncertainty is an open research topic.

**Table 2.** Accuracy measures (the mean and standard deviation) for comparison of the well-known MIL classification models, the RF and the STE-MIL using datasets Elephant, For and Tiger.

|  | **Elephant** | **Fox** | **Tiger** |
| --- | --- | --- | --- |
| mi-SVM [15] | $0.822 \pm$ N/A | $0.582 \pm$ N/A | $0.784 \pm$ N/A |
| MI-SVM [15] | $0.843 \pm$ N/A | $0.578 \pm$ N/A | $0.840 \pm$ N/A |
| MI-Kernel [54] | $0.843 \pm$ N/A | $0.603 \pm$ N/A | $0.842 \pm$ N/A |
| EM-DD [55] | $0.771 \pm 0.097$ | $0.609 \pm 0.101$ | $0.730 \pm 0.096$ |
| mi-Graph [56] | $0.869 \pm 0.078$ | $0.620 \pm 0.098$ | $0.860 \pm 0.083$ |
| miVLAD [57] | $0.850 \pm 0.080$ | $0.620 \pm 0.098$ | $0.811 \pm 0.087$ |
| miFV [57] | $0.852 \pm 0.081$ | $0.621 \pm 0.109$ | $0.813 \pm 0.083$ |
| mi-Net [19] | $0.858 \pm 0.083$ | $0.613 \pm 0.078$ | $0.824 \pm 0.076$ |
| MI-Net [19] | $0.862 \pm 0.077$ | $0.622 \pm 0.084$ | $0.830 \pm 0.072$ |
| MI-Net with DS [19] | $0.872 \pm 0.072$ | $0.630 \pm 0.080$ | $0.845 \pm 0.087$ |
| MI-Net with RC [19] | $0.857 \pm 0.089$ | $0.619 \pm 0.104$ | $0.836 \pm 0.083$ |
| Attention [23] | $0.868 \pm 0.022$ | $0.615 \pm 0.043$ | $0.839 \pm 0.022$ |
| Gated-Attention [23] | $0.857 \pm 0.027$ | $0.603 \pm 0.029$ | $0.845 \pm 0.018$ |
| **STE-MIL** | $\mathbf{0.885} \pm 0.038$ | $\mathbf{0.730} \pm 0.080$ | $\mathbf{0.875} \pm 0.039$ |

**Table 3.** Accuracy measures (the mean and standard deviation) for comparison of the well-known MIL classification models, the RF and the STE-MIL by using datasets Musk1 and Musk2.

|  | **Musk1** | **Musk2** |
| --- | --- | --- |
| mi-SVM [15] | $0.874 \pm$ N/A | $0.836 \pm$ N/A |
| MI-SVM [15] | $0.779 \pm$ N/A | $0.843 \pm$ N/A |
| MI-Kernel [54] | $0.880 \pm$ N/A | $0.893 \pm$ N/A |
| EM-DD [55] | $0.849 \pm 0.098$ | $0.869 \pm 0.108$ |
| mi-Graph [56] | $0.889 \pm 0.073$ | $\mathbf{0.903} \pm 0.086$ |
| miVLAD [57] | $0.871 \pm 0.098$ | $0.872 \pm 0.095$ |
| miFV [57] | $0.909 \pm 0.089$ | $0.884 \pm 0.094$ |
| mi-Net [19] | $0.889 \pm 0.088$ | $0.858 \pm 0.110$ |
| MI-Net [19] | $0.887 \pm 0.091$ | $0.859 \pm 0.102$ |
| MI-Net with DS [19] | $0.894 \pm 0.093$ | $0.874 \pm 0.097$ |
| MI-Net with RC [19] | $0.898 \pm 0.097$ | $0.873 \pm 0.098$ |
| Attention [23] | $0.892 \pm 0.040$ | $0.858 \pm 0.048$ |
| Gated-Attention [23] | $0.900 \pm 0.050$ | $0.863 \pm 0.042$ |
| **STE-MIL** | $\mathbf{0.918} \pm 0.077$ | $0.854 \pm 0.061$ |

## 8. Conclusions

### 8.1. Discussion

An RF-based model to solve the MIL classification problem for small tabular data is proposed. It is based on training decision trees by means of their converting to a neural network of a specific form. Moreover, it uses the attention mechanism to aggregate the bag

information and to enhance the classification accuracy. The attention mechanism can also be used to explain why a tested bag is assigned a certain label, because the attention shows weights of instances of the tested bag and selects the most influential instances.

Numerical experiments with well-known datasets, used by many authors in evaluating MIL models, demonstrated that STE-MIL outperformed many models, including the following: mi-SVM, MI-SVM, MI-Kernel, EM-DD, mi-Graph, miVLAD, miFV, mi-Net, MI-Net, MI-Net with DS, MI-Net with RC, the Attention and Gated-Attention models.

An important advantage of STE-MIL is that it successfully combines the positive properties of decision trees to accurately classify small tabular data, with properties of neural networks to learn complex functions in an end-to-end manner. This combination was implemented by introducing soft decision trees and converting the decision trees into neural networks. Another beneficial idea behind STE-MIL was that of using the tree outputs, in the form of the class probability distributions, as embeddings, which allowed us to apply the attention mechanism. Moreover, we propose several additional improvements which enhance the classification accuracy of the whole MIL model. These include the use of axis-parallel separating hyperplanes in building the decision trees, the use of the softmax operation as an approximation of the argmax operation, the original method to transform the decision tree to a neural network.

The proposed ideas and improvements made it possible to create a fairly effective tool to solve the problem of MIL under the condition of having a small amount of tabular training data.

### 8.2. Open Research Questions

There are several open research questions to study in order to significantly improve the proposed STE-MIL. Moreover, ideas behind STE-MIL can be used in other known MIL models.

First, it is interesting to study how neighboring patches or instances of each analyzed patch can be incorporated into the STE-MIL scheme, as in [25]. The incorporation of neighbors can significantly improve STE-MIL and enhance its classification accuracy.

It should be noted that RF, as an ensemble of decision trees, was used in STE-MIL. At the same time, the gradient boosting machine [59,60] is also an efficient model, which uses decision trees as weak learners and can also be used in the STE-MIL scheme. However, we meet several open questions in the use of a gradient boosting machine. First, it is not obvious how to implement the attention mechanism in this case. The problem is that each tree in the gradient boosting is built on a new dataset consisting of residuals. Second, the end-to-end learning of the whole model is also an open question.

Another open question is how to adapt the proposed model to large image data; for example, in the case of histology images. In this case, we need to reduce images to tabular data in order to build decision trees. One of the ways to do so is to use the autoencoder for each image instance to get the corresponding embedding of the low dimension. The question is how to incorporate this autoencoder in the STE-MIL scheme to train in and end-to-end manner.

The above questions can be regarded as directions for further research to improve the MIL model.

### 8.3. Concluding Remarks

An important peculiarity of STE-MIL is that it opens a door for the construction of various models which use trainable decision trees as neural networks. In contrast to models using oblique decision trees, the proposed trainable trees have a significantly small number of training parameters, preventing overfitting of the training process. Therefore, these models could be effective when small tabular datasets are considered. Another peculiarity of STE-MIL is that the model is very simple and clear. All components of the model are simply implemented.

We have shown how the introduced components of STE-MIL, including the soft tree ensemble, the transformation procedure of decision trees to neural networks, and the representation of the predicted class probability distributions produced by the trees as embeddings, can be used in MIL models to obtain outperforming results. However, they can be applied to a wide range of machine learning models and tasks which aim to classify instances based on small tabular datasets. Therefore, the contribution of the components proposed in STE-MIL goes beyond the application considered in the work.

**Author Contributions:** Conceptualization, L.U. and A.K.; methodology, L.U. and V.M.; software, A.K.; validation, V.M. and A.K.; formal analysis, A.K. and L.U.; investigation, A.K. and V.M.; resources, L.U. and V.M.; data curation, V.M.; writing—original draft preparation, L.U. and A.K.; writing—review and editing, A.K. and V.M.; visualization, A.K.; supervision, L.U.; project administration, V.M.; funding acquisition, V.M. All authors have read and agreed to the published version of the manuscript.

**Funding:** The research is partially funded by the Ministry of Science and Higher Education of the Russian Federation as part of the World-class Research Center program: Advanced Digital Technologies (contract No. 075-15-2022-311 dated 20 April 2022).

**Data Availability Statement:** Not applicable.

**Acknowledgments:** The authors would like to express their appreciation to the anonymous referees whose very valuable comments have improved the paper.

**Conflicts of Interest:** The authors declare no conflict of interest.

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
