# Peer review of "Multiple Instance Learning with Trainable Soft Decision Tree Ensembles"

_algorithms, doi:10.3390/a16080358_

Round 1
Reviewer 1 Report
Please check attachment.

Moderate editing of English language required.
Reviewer 2 Report
This manuscript [generally] reads well [see my comments below] and is [again generally] logically structured with a clear narrative. I noted the availability of the program code (requires gitHub) and the source of the datasets which will be useful for other research groups to test for reproducibility and re-use as noted in the manuscript which has in the past proved to possible in approximately 10% of published research as evidenced in academic publications.
The manuscript has a descriptive title and keywords Iindex terms) howeverm the abstract, while adequate as a summary, should not include acronyms which should be restricted to the main body text of the manuscript and defined on first use.
I have a number of comments:
1) While the paper generally reads well thete are multiple (albeit minor) errors in the use of English grammar. The errors are principally in the use of the indefinite article and plurals plus (e.g., the word selection - the word "sum" should be 'summary'). Careful proof checking is required.
2) The conclusion is in reality a conflation of a discussions section, future work, and closing observations. This section needs to be rewritten and revised (extended) inot three dedicated sections: (i) a discussion where the threads seto out in the syudy are drawn together and discussed along with the results and the conclusions drawn from the results, (ii) a section where open research questions (ORQ) are consodered along with future directions for research (the statement in the current conclusion "Therefore, another idea behind new models is to apply the gradient boosting machine. This is another direction for further research.") fails to address this, and (iii) a section setting out the concluding observations.
3) I appreciate the motivation for this research but the authors need to provide more consideration of the practical managerial significance of their work and proposed solution long with possibly illustrative practical applications (a feature of curent practice which fuses original research with practical application in a fusion approach) which are not adequately adderssed in the manuscript.
4) I noted in table 2: Gated-Attention [23] 0.857±0.027 0.603±0.029 0.845±0.018 STE-MIL 0.885±0.038 0.730±0.080 0.875±0.039 and intable 3: Attention [23] 0.892±0.040 0.858±0.048 Gated-Attention [23] 0.900±0.050 0.863±0.042 STE-MIL 0.918±0.077 0.854±0.061. In these results the SD for [23] appears to be better than for the proposed solution. The authors may wish to discuss further the results (including any ORQ) relating to this point.
In summary, I found this to generally to be a good paper which, while relatively specialised, will be of interest to the journal and the potential audience. However, there are areas where revision is required as set out in my review. When suitably revised the paper will, in my view, be a suitable candidate for publication.
Round 2
Reviewer 1 Report
The authors need to do track changes to the article. It is very difficult to know where the correction have been made in the manuscript.
English needs improvement as some sentences do not make sense. iT needs to go through proof reading be a native English speaker.
Reviewer 2 Report
I an content with the revisions which in my view have addressed the reviewer comments.